

# Exploring differential gene expression and biomarker potential in systemic lupus erythematosus: a retrospective study

Jie Xiao[1,2,*], Yuhong Luo[1,2,*], Lina Duan[1,2], Xinru Mao[1,2], Lingyue Jin[1,2], Haifang Wang[1,2], Hongxia Wang[1,2], Jie Pan[1,2], Ying Gong[1,2,3] and Haixia Li[1,2]

[1] Department of Laboratory Medicine, Guangdong Provincial Key Laboratory of Precision Medical Diagnostics, Guangdong Engineering and Technology Research Center for Rapid Diagnostic Biosensors, Guangdong Provincial Key Laboratory of Single-cell and Extracellular Vesicles, Nanfang Hospital, Southern Medical University, Guangzhou, PR China
[2] Guangdong Provincial Clinical Research Center for Laboratory Medicine, Guangzhou, Guangdong, China
[3] Department of Internal Medicine, Division of Hematology, University of Maastricht, Maastricht, Netherlands
[*] These authors contributed equally to this work.

Corresponding authors
Ying Gong,
gongy3@mail2.sysu.edu.cn
Haixia Li, yingchun1220@163.com

## ABSTRACT

**Background**. Systemic lupus erythematosus (SLE) is a chronic autoimmune disease characterized by inflammation and immune-mediated multi-organ system damage, accompanied by clinical manifestations such as fever, hair loss, skin rash, oral ulcers, and joint pain and swelling. SLE has been reported to affect more than 3.4 million people worldwide, of which approximately 90% are women.

**Purpose**. This study aims to identify and characterize key hub genes implicated in SLE through comprehensive bioinformatics analyses, providing a theoretical foundation for the development of more effective therapeutic strategies.

**Methods**. Two datasets were procured from the Gene Expression Omnibus (GEO) database: GSE13887 and GSE10325. Differentially expressed genes (DEGs) were identified and subjected to functional enrichment analysis, protein-protein interaction (PPI) network construction, and receiver operating characteristic (ROC) curve analysis to evaluate potential hub genes. The top 20 significantly upregulated and downregulated DEGs, alongside the top 15 enriched Gene Ontology (GO) terms and five Kyoto Encyclopedia of Genes and Genomes (KEGG) pathways, were screened from both datasets. Quantitative real-time PCR (RT-q PCR) was utilized to validate hub gene expression in CD3 + T cells from peripheral blood samples of SLE patients. Concurrently, flow cytometry was employed to quantify inflammatory cytokines in peripheral blood samples.

**Results**. Bioinformatics analyses identified 1,912 DEGs in GSE13887 and 52 DEGs in GSE10325, with eight DEGs common to both datasets. Functional enrichment analysis underscored critical biological processes, notably cell-mediated cytotoxicity and cell killing. PPI network and enrichment analyses highlighted seven hub genes, among which *FCER1A* and *RGS1* demonstrated consistent expression trends across datasets and clinical samples—*FCER1A* was significantly downregulated, while *RGS1* was upregulated in SLE patients. ROC curve analysis confirmed their strong diagnostic potential (AUC > 0.7). Principal component analysis (PCA) further highlighted distinct gene expression profiles differentiating SLE patients from healthy controls.

Clinical validation *via* RT-q PCR and flow cytometry corroborated these findings, demonstrating decreased FCER1A expression and increased RGS1 expression in CD3 + T cells from SLE patients. Moreover, elevated plasma levels of IL-6 and TNF-α, coupled with diminished IL-10 levels, were observed in SLE patients. These findings suggest that FCER1A and RGS1 are promising biomarkers for SLE diagnosis.

**Conclusions**. FCER1A and RGS1 are significantly associated with SLE and serve as potential biomarkers for distinguishing SLE patients from healthy individuals. Their involvement in SLE pathogenesis underscores their potential as targets for future diagnostic and therapeutic interventions.

# INTRODUCTION

Systemic lupus erythematosus (SLE) is a chronic autoimmune disorder characterized by multisystemic involvement, affecting various organs such as the skin, kidneys, joints, and central nervous system (*Hoi et al., 2024*). It impacts approximately 3.4 million individuals globally (*Tian et al., 2023*) and predominantly affects women of childbearing age, with a female-to-male ratio of approximately 10:1 (*Fanouriakis et al., 2021*). In recent decades, the incidence of SLE has shown a continuous upward trend (*Barber et al., 2021*; *Gergianaki et al., 2017*). The global prevalence of SLE varies across regions, ranging from 20 to 150 cases per 100,000 people. The disease burden is notably higher among certain racial and ethnic groups, including Black, Asian, and Hispanic populations (*Barber et al., 2021*; *Siegel & Sammaritano, 2024*).

The pathogenesis of SLE is intricate, involving a multifactorial interplay of genetic, epigenetic, environmental, and hormonal factors (*Tsokos, 2024*; *Tsokos, 2011*). Its pathogenesis is involved in the alteration of whole immune system. Dysregulated T cell responses are pivotal in the immunopathology of SLE, with CD4$^+$ T helper (Th) cells, particularly Th1, Th2, Th17, and regulatory T cells (Tregs), playing crucial roles in disease progression (*Sharabi & Tsokos, 2020*). An imbalance in Th1/Th2 cytokine profiles and the heightened presence of proinflammatory cytokines, such as interleukin 6 (IL-6), interleukin 17 (IL-17), and interferon-α (IFN-α), are strongly associated with disease activity and organ damage (*Li et al., 2022*). Moreover, impaired Treg function exacerbates immune dysregulation and fosters the generation of pathogenic autoantibodies (*Brusko, Putnam & Bluestone, 2008*). B cell hyperactivity and the subsequent overproduction of autoantibodies, such as anti-dsDNA and anti-Smith (Sm) antibodies, are defining features of SLE (*Pisetsky, Bossuyt & Meroni, 2019*; *Karrar & Cunninghame Graham, 2018*). These autoantibodies form immune complexes that are deposited in various tissues, instigating complement activation, inflammation, and tissue damage (*Tsokos et al., 2016*). Furthermore, neutrophil extracellular traps (NETs) have been implicated in the pathogenesis of SLE by exacerbating

self-antigen exposure and perpetuating type I interferon expression (*Papayannopoulos, 2018*).

Despite significant advances in the understanding of disease mechanisms, the early diagnosis and management of SLE remain challenging due to its heterogeneous clinical manifestations and unpredictable progression (*Dörner & Furie, 2019*; *Durcan, O'Dwyer & Petri, 2019*). Current therapeutic approaches primarily target immunosuppression and symptom alleviation; however, these strategies are frequently accompanied by substantial side effects and variable therapeutic efficacy (*Fanouriakis et al., 2019*). The diagnosis of SLE is predominantly based on clinical presentation and laboratory testing. The American College of Rheumatology (ACR) and the Systemic Lupus International Collaborating Clinics (SLICC) have established classification criteria, which incorporate both clinical manifestations (*e.g.*, cutaneous rashes, joint involvement, renal dysfunction) and serological markers, including the presence of antinuclear antibodies (ANA), anti-dsDNA antibodies, Sm antibodies, and reduced complement levels (*Styrkarsdottir et al., 2021*). Additional diagnostic modalities include urinalysis to assess renal involvement and skin biopsies in cases of cutaneous lupus.

However, these diagnostic methods are not without their limitations. First, the clinical manifestations of SLE are highly heterogeneous and may overlap with those of other autoimmune diseases, rendering early diagnosis challenging (*Dörner & Furie, 2019*). For instance, symptoms such as fatigue, arthralgia, and rashes are not exclusive to SLE and may occur in a variety of other conditions, leading to delayed identification and intervention. Secondly, the presence of autoantibodies such as ANA lacks specificity for SLE, as these antibodies may also be found in other autoimmune disorders or even in healthy individuals (*Durcan, O'Dwyer & Petri, 2019*). Furthermore, some patients with SLE may test negative for certain autoantibodies, complicating the diagnostic process. Lastly, these diagnostic methods often fail to adequately capture disease activity or predict organ damage, thus limiting their effectiveness in monitoring disease progression and guiding therapeutic strategies (*Fanouriakis et al., 2019*). Traditional diagnostic approaches are time-intensive and cumbersome, hindering their utility in the timely diagnosis, treatment, and prognosis of patients.

Genetic testing has increasingly become a pivotal tool in the diagnosis and treatment of SLE, significantly enhancing both the accuracy and efficiency of diagnosis and therapeutic strategies. By identifying genetic biomarkers linked to SLE, these tests facilitate earlier and more precise diagnoses, thereby enabling timely interventions and optimizing patient management. Consequently, genetic screening is now an essential component of clinical practice, not only refining diagnostic precision but also paving the way for more personalized treatment approaches. Through the identification of disease-associated genes *via* genetic testing, coupled with the validation of corresponding biomarkers through techniques such as flow cytometry, SLE can be diagnosed and managed more swiftly and effectively, allowing for a more targeted approach to treatment. This, in turn, improves both short-term clinical outcomes and long-term disease control (*Ghodke-Puranik, Olferiev & Crow, 2024*; *Vasquez-Canizares, Wahezi & Putterman, 2017*; *Lee et al., 2022*; *Horisberger et al., 2022*; *Zhu et al., 2023*).

**Table 1 Characteristics of SLE patients and HCs.** This table details the characteristics of patients with SLE and HC. Data are presented as mean (minimum–maximum) for continuous variables and n (%) for categorical variables.

| General information | | SLE ($n = 45$) | HC ($n = 40$) | $p$ value |
|---|---|---|---|---|
| gender | Female | 42 (93.33%) | 39 (97.5%) | 0.368 |
| | Male | 3 (6.67%) | 1 (2.5%) | |
| Age, years | | 38.09 (15–74) | 42.20 (24–67) | 0.099 |
| WBC, $10^9$/L | | 6.72 (1.96–21.97) | 6.00 (3.65–9.54) | 0.695 |
| LYM, $10^9$/L | | 1.74 (0.07–3.92) | 2.07 (1.38–3.04) | 0.008[2] |
| NEU, $10^9$L | | 4.36 (1.12–17.84) | 4.63 (1.72–52.80) | 0.269 |
| RBC, $10^{12}$/L | | 4.29 (2.46–5.74) | 4.62 (3.64–5.18) | 0.003[2] |
| HGB, g/L | | 124.71 (78–163) | 134.80 (99–165) | 0.003[2] |
| HCT, L/L | | 0.38 (0.25–0.48) | 0.41 (0.33–0.49) | 0.002[2] |
| PLT, $10^9$/L | | 225.09 (9–377) | 260.78 (144–387) | 0.051 |
| SLEDAI | | 9.44 (1–24) | —— | —— |

**Notes.**

Abbreviations: WBC, white blood cell; LYM, lymphocyte; NEU, neutrophil; RBC, red blood cell; HGB, hemoglobin; HCT, hematocrit; PLT, platelet.

Data represent mean values (minimum-maximum values). $P$ values were calculated by Mann–Whitney $U$ test or $t$ test.

[1]$p < 0.05$.
[2]$p < 0.01$.
[3]$p < 0.001$.

The purpose of this study is to explore genes, biomarkers and molecular processes related to SLE in combination with comprehensive bioinformatics analysis, and verify them through clinical data to better diagnose SLE.

# MATERIALS AND METHODS

## Research participants and sample collection

This study is a retrospective study. A total of 45 SLE patients and 40 healthy controls (HCs) were collected from Nanfang Hospital of Southern Medical University from January 2024 to December 2024. All SLE patients met the 2012 ACR SLE disease classification criteria (*Petri et al., 2012*; *Tan et al., 1982*). All the SLE patients with concurrent infection, acute concurrent illness, and use of probiotics or antibiotics within 1 month before admission were excluded. The collected HCs also needed to have no history of known autoimmune diseases and be of comparable age to SLE patients. All the participants were collected two mL EDTA-$K_2$ blood sample.

Ethics approval was granted by the Ethics Committee of Nanfang Hospital, under the surveillance of ethical number: NFEC-2025-026. All the methods used were in accordance with the approved guidelines. Written informed consent was required from all patients and healthy volunteers in the study.

The SLE patients and HCs in this study were comparable in age and gender (Table 1). Average age of the SLE and HC groups was 38.09 ± 11.97 and 42.20 ± 10.34, respectively ($p = 0.099$).

## Datasets

Two raw datasets, GSE13887 and GSE10325, were downloaded from the Gene Expression Omnibus (GEO) database (https://www.ncbi.nlm.nih.gov/geo/). GSE13887 was an analysis of CD3-positive T cells isolated from 10 SLE patients and nine HCs. GSE10325 was an analysis of freshly isolated lymphocytes (CD4$^+$ T cells and CD19$^+$ B cells) and CD33$^+$ myeloid subsets from the blood of 28 HCs and 39 SLE patients.

## Identification of differentially expressed genes

Differentially expressed genes (DEGs) between SLE and HCs were identified using the *limma* package (version 3.64.1) in R (version 4.2.1). The GSE13887 and GSE10325 datasets were first converted into expression matrices, appropriately grouped, and normalized. Normalization and differential expression analysis were performed using the *limma* package. Genes with an adjusted *p*-value (false discovery rate, FDR) < 0.05 and an absolute log2 fold change (|log2FC|) $\geq$ 0.5 were considered statistically significant. DEGs were classified as upregulated or downregulated based on whether their log2FC values were above 0.5 or below −0.5, respectively. An online Venn diagram tool was employed to identify overlapping DEGs between SLE and HC. The results were visualized using *ggplot2* (version 3.4.4) for volcano plots (with thresholds set at $p \leq 0.05$ and |log2FC| $\geq 1$) and box plots to assess normalization across datasets. Heatmaps were generated using the *ComplexHeatmap* package (version 2.13.1), while unique and shared DEGs were further illustrated using the *VennDiagram* package (version 1.7.3) (*Gu, Eils & Schlesner, 2016*).

## Gene Ontology term and Kyoto Encyclopedia of Genes and Genomes (KEGG) pathway enrichment analysis

The R language packages clusterProfiler (version 4.4.4) and ggplot2 were used to perform Gene Ontology (GO) term and Kyoto Encyclopedia of Genes and Genomes (KEGG) pathway enrichment analysis on common DEGs (co-DEGs) and visualize the results, with a threshold of *p* value < 0.05, and bar charts, bubble charts, and chord plots were drawn (*Yu et al. , 2012*).

## Construction of protein-protein interaction network and identification of hub genes

The PPI network of DEGs that may play an important role in the progression of SLE was constructed using the STRING online database (https://www.string-db.org/). The protein–protein interaction (PPI) network was performed using Cytoscape (version 3.10.2) (http://www.cytoscape.org/) to better visualize the interaction information. Hub genes with connectivity $\geq$10 were selected using the CytoHubba (version 0.1).

## Hub gene validation

Receiver operating characteristic (ROC) analysis was performed on the data using the R language package pROC (version 1.18.0). The results were visualized using ggplot2 to plot the ROC curve of the hub gene. The area under the curve (AUC) of the ROC curve corresponding to the hub gene was used to evaluate the ability to distinguish between SLE and HC. The expression profiles of the hub genes in the two data sets were used as variables

for principal component analysis (PCA) to obtain PC1 and PC2, and ggplot2 was used to visualize the results.

## CD3$^+$ T cell isolation from SLE and HC periphal blood samples

Human primary PBMC were isolated from periphral blood with EDTA-K2 by density gradient centrifugation using Ficoll. 1 X 10$^6$ CD3$^+$ T cells were enriched by positive selection with an CD3$^+$ T cells isolation kit (130-097-043, Miltenyi Biotec, Bergisch Gladbach, Germany). The purity of T cell were analyzed by Flow cytometry (CantoII, BD Bioscience, Franklin Lakes, NJ, USA) staining with anti-CD3 -PE antibody (562310, clone: HIT3A, BD Bioscience, Franklin Lakes, NJ, USA).

## Real-time quantitative polymerase chain reaction

The sorted CD3$^+$ T cells were lyzed with TRIzol reagent (Cat. #G3013 Servicebio, Wuhan, China). After chloroform substitute (Cat. #G3014 Servicebio, Wuhan, China) isolation, RNA were percipitated by isopropanol by centrifugation at 4 °C, 15,000 g, 15 min. After that, RNA were rinsed with 2 times 75% ethanol. The cDNA were synthesized with SweScript RT I First Strand cDNA Synthesis Kit (Cat. # G331-1, Servicebio, Wuhan, China) by Random Hexamer Primer (Cat. #G331-4, Servicebio, Wuhan, China). The RNA template were added one µg.

The qPCR were conducted by using Servicebio$^®$ 2×SYBR Green qPCR Master Mix (Cat. #G3326-1, Servicebio, Wuhan, China). The reation volume were used 20 µL in each wells, cDNA templated were diluted in 10 times with DEPC water and 2 µL, forward and reverse primers were added 0.4 µL. The qPCR were performed with Roche Cobas 4800 PCR machine (Roche, Basel, Switzerland). This experiment used a two-step reaction procedure: pre-denaturation: 95 °C, 2–5 min; cyclic amplification (35–45 cycles): 95 °C denaturation 5–15 s; 60–68 °C annealing/extension 20–60 s (simultaneous fluorescence collection, SYBR Green collection at the end point, TaqMan collection in real time). Melting curve (SYBR Green): gradually increase the temperature from 65 °C to 95 °C and monitor the fluorescence. Relative gene expression was calculated using the $2^{-\Delta\Delta Ct}$ method, and the results were normalized using *GAPDH*. Primer sequences are shown in Table S1.

## Flow cytometry

The FCER1A and RGS1 protein expression levels of CD3$^+$ T cells in plasma collected from HCs and SLE patients and the levels of inflammatory factors in plasma were detected by flow cytometry. Samples were analyzed using a BD FACS Fortassa flow cytometer (BD Biosciences, USA). Data acquisition was performed using BD FACSDiva software, and at least 10,000 events were recorded for each sample. Polyclonal rabbit anti-human RGS1 antibody (Cat# LS-C162570-400; LSBio, Seattle, WA, USA) and FITC-conjugated goat-anti-rabbit secondary antibodies were used for RGS1 analysis (*Jiang et al., 2024*). Polyclonal rabbit anti-human RGS1 antibody (Cat# LS-C162570-400; LSBio, Seattle, WA, USA) and FITC-conjugated goat-anti-rabbit secondary antibodies were used for RGS1 analysis. A rabbit polyclonal FCER1 alpha Monoclonal Antibody (Cat# 16-5899-82, Functional Grade, eBioscience, San Diego, CA, USA) was used for FCER1A analysis (*Greer et al., 2014*).

## Statistics

All data processing and analysis were performed in GraphPad Prism 9.0. To compare two groups of variables, the Mann–Whitney U test was applied. Fisher's exact test was performed to analyze the statistical significance between two variable data sets. $P < 0.05$ was considered statistically significant. Statistical significance is reported as follows: $p < 0.05(*)$, $p < 0.01(**)$, $p < 0.001(***)$, $p < 0.0001(****)$, ns(not significant).

## RESULTS

### Identification of DEGs

Utilizing bioinformatics analysis through R software (version 4.2.1), a total of 1,912 DEGs were identified in the GSE13887 dataset, while 52 DEGs were detected in the GSE10325 dataset. Subsequent hierarchical clustering analysis was performed on these DEGs, prioritizing the top 20 upregulated and downregulated genes from GSE13887 (Table S3) and GSE10325 (Table S4), respectively. Volcano plots and heatmaps were generated to visualize the clustering results for GSE13887 (Figs. 1A, 1C) and GSE10325 (Figs. 1B, 1D). The heatmaps distinctly illustrated a high degree of consistency in sample clustering.

Following data normalization and comparative evaluation across datasets, box plots were constructed to confirm the normalization quality for GSE13887 (Fig. 1E) and GSE10325 (Fig. 1F). The results demonstrated that the data distribution across both datasets conformed to established quality standards, affirming the robustness and cross-comparability of the microarray data. A Venn diagram (Fig. 1G) was subsequently generated to illustrate the overlap of DEGs between the two datasets, identifying eight shared DEGs.

### Enrichment analysis of DEGs

GO and KEGG pathway enrichment analyses were conducted to elucidate the functional roles of the eight common differentially expressed genes (co-DEGs). GO terms were categorized into biological process (BP), cellular component (CC), and molecular function (MF), with the top five most significantly enriched terms in each category selected based on the lowest *p*-values (Table S5). These results were visualized *via* bar charts (Fig. 2A) and bubble plots (Fig. S1A). The co-DEGs were predominantly associated with processes such as *cell killing, leukocyte-mediated cytotoxicity, natural killer cell-mediated immunity,* and *natural killer cell-mediated cytotoxicity.*

Subsequently, the five most significantly enriched KEGG pathways were identified (Table S6) and represented through bar plots (Fig. 2B) and bubble charts (Fig. S1B). The co-DEGs were chiefly involved in pathways related to *transcriptional dysregulation in cancer, apoptosis, thyroid cancer, African trypanosomiasis,* and *asthma.*

### PPI network and hub genes

A chord diagram was generated to illustrate the enrichment of genes within the top five GO categories across BP, CC, and MF, identifying seven key genes: GZMB, LAG3, APOL1, CXCL13, RGS1, FCER1A, and DPEP2. Notably, RGS1, CXCL13, and GZMB were enriched in at least two GO categories (Figs. 2C–2E). The DEGs were uploaded to the

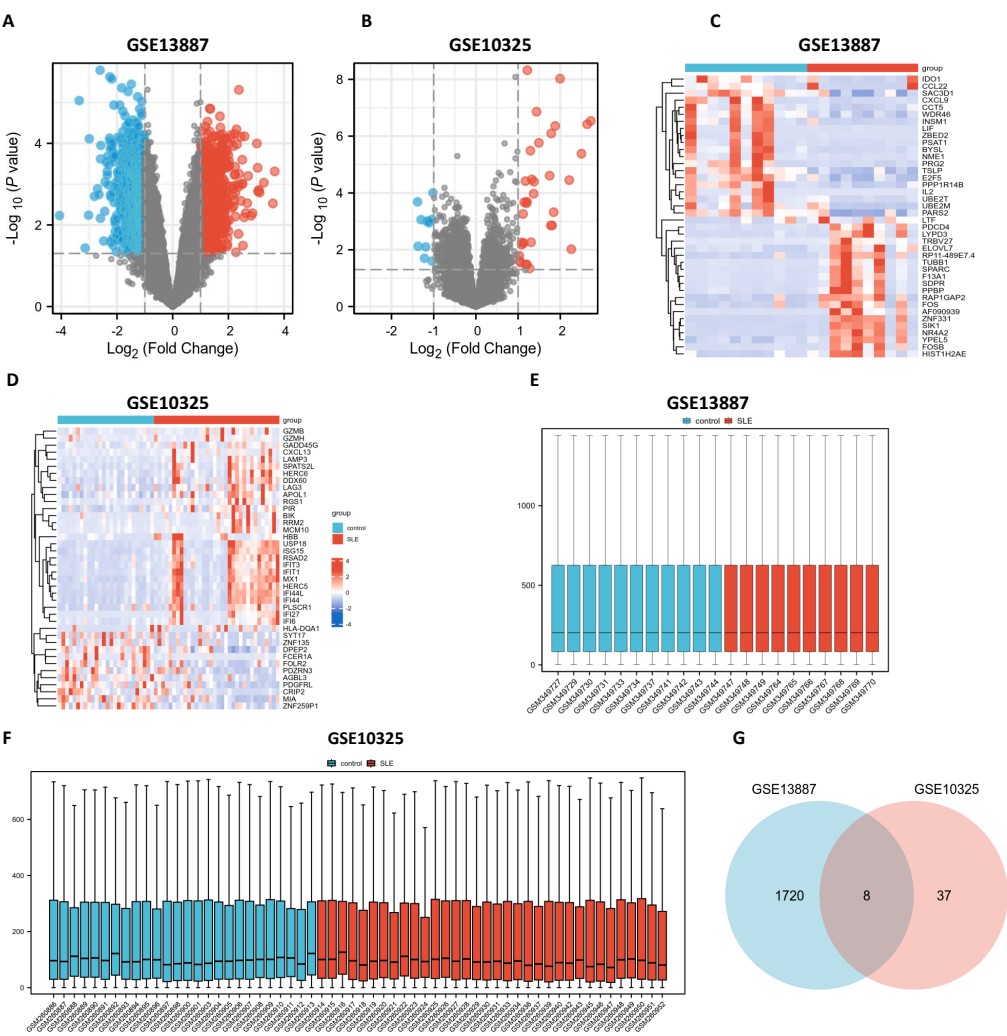

**Figure 1  Volcano plots, heat maps, box plots, and a Venn diagram of GSE13887 and GSE10325 datasets.** (A) Volcano plot of GSE13887 dataset. The *x*-axis represents log2 (Fold Change), and the *y*-axis represents −log10 (*P* value). Red dots represent upregulated genes, and blue dots represent downregulated genes. (B) Volcano plot of GSE10325 dataset. (C) Heat map of the top 20 upregulated and downregulated EP-DEGs in GSE13887 dataset. Each row represents a gene, and each column represents a sample. Red represents high expression levels, and blue represents low expression levels. (D) Heat map of GSE10325 dataset. (E) Box plot of GSE13887 dataset, with no significant difference in median and upper and lower quartiles. (F) Box plot of GSE10325 dataset. (G) Venn diagram of common DEGs in GSE13887 and GSE10325 datasets.

STRING database to construct the PPI network (Fig. S2). Using the MCODE plugin, a tightly interconnected protein cluster consisting of CXCL13, GZMB, LAG3, and FCER1A was identified (Fig. 3A).

Overall, seven hub genes—GZMB, LAG3, APOL1, CXCL13, RGS1, FCER1A, and DPEP2—were selected based on both the PPI network and chord diagram analyses. Volcano plot assessments of their expression levels in the GSE13887 and GSE10325 datasets revealed that only FCER1A and RGS1 exhibited consistent expression trends across both datasets,

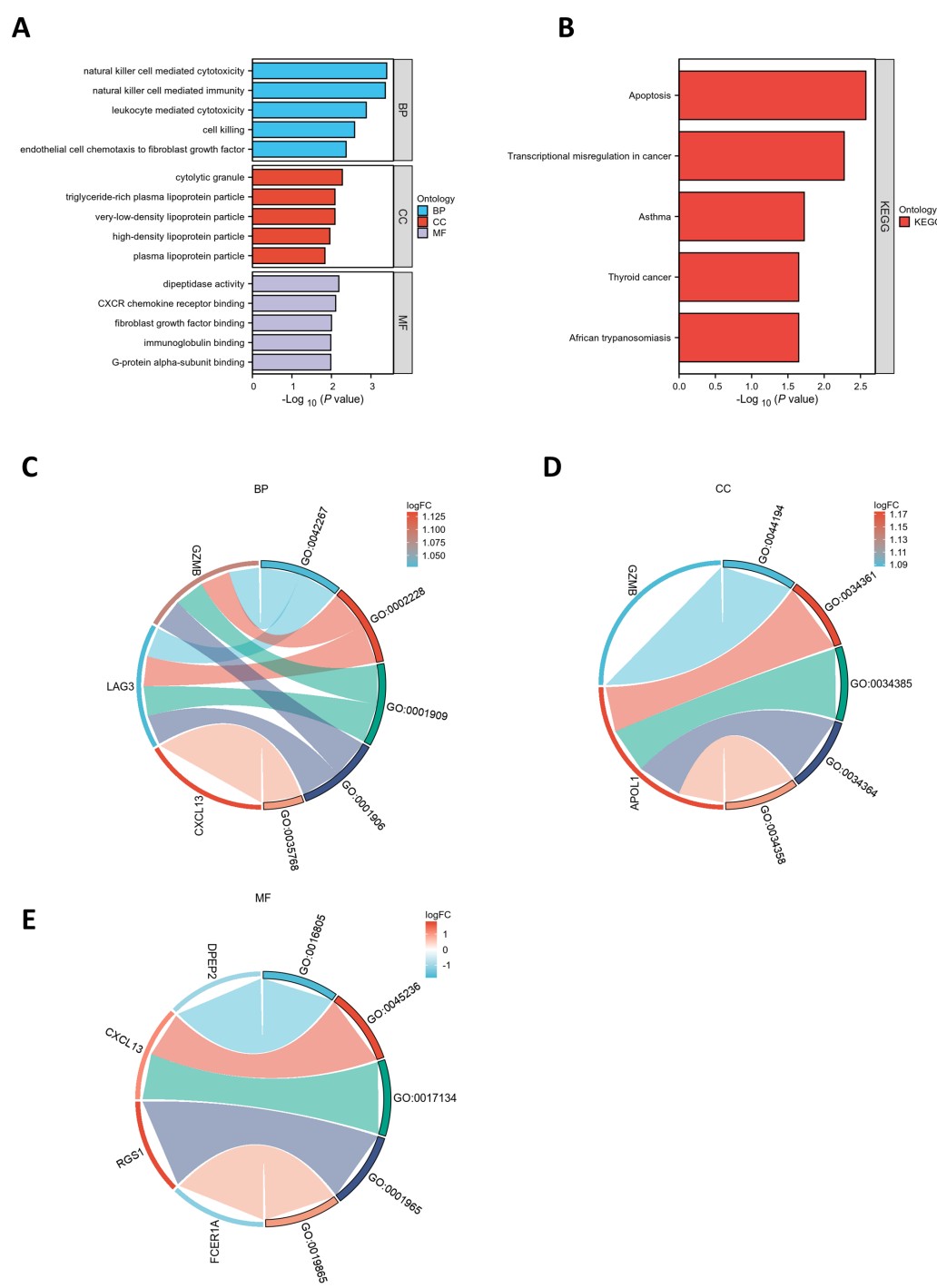

**Figure 2 Functional enrichment analysis of DEGs.** (A) Bar plot of GO enrichment analysis results for DEGs. The *x*-axis represents GO terms, while the *y*-axis displays −log10 (*p*-value) for each term. (B) Bar plot of KEGG pathway enrichment analysis results. (C-E) Chord diagrams depicting EP-DEGs enriched in the top five GO categories of biological processes (BP), cellular components (CC), and molecular functions (MF). Red indicates upregulation, blue denotes downregulation, and color intensity corresponds to the magnitude of expression changes.

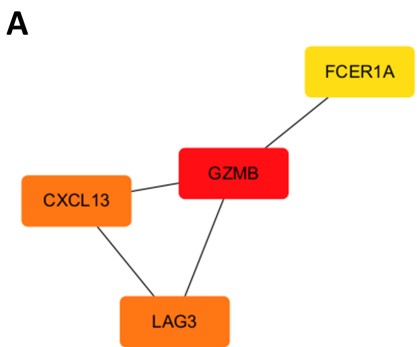

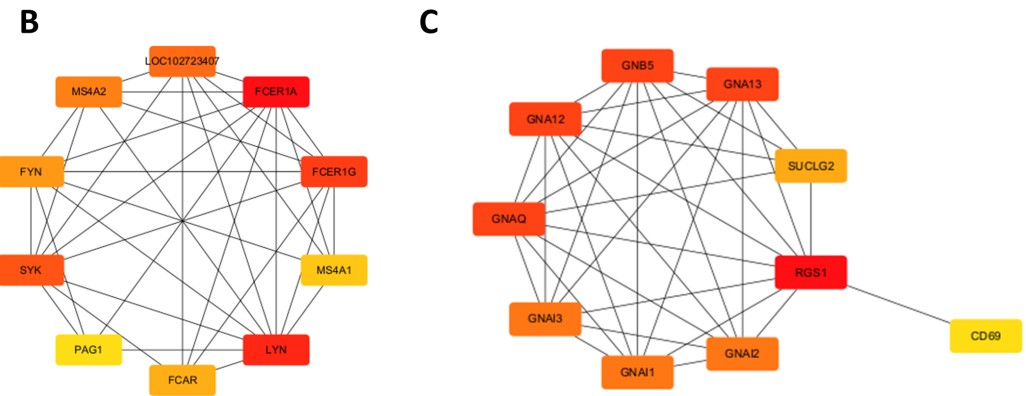

**Figure 3 Identification of hub genes in the PPI network and functional enrichment analysis of** ***FCER1A*** **and** ***RGS1***. (A) Identification of four hub genes (*CXCL13, GZMB, LAG3, FCER1A*) using CytoHubba in Cytoscape. (B) Top ten hub genes identified in the *FCER1A* PPI network using CytoHubba in Cytoscape. (C) Top ten hub genes identified in the *RGS1* PPI network using CytoHubba in Cytoscape.

with FCER1A being downregulated and RGS1 upregulated (Fig. S3). These two genes were further analyzed using the STRING database to construct an additional PPI network (Fig. S4), visualized in Cytoscape. MCODE analysis identified a highly interconnected protein cluster comprising ten genes (Figs. 3B, 3C).

## Prognostic value of hub genes

ROC curves were generated to evaluate the diagnostic efficacy of *FCER1A* and *RGS1*, both individually and in combination, using their expression levels in the GSE13887 and GSE10325 datasets (Fig. 4). Both genes demonstrated strong diagnostic potential, with area under the curve (AUC) values exceeding 0.7. The combined analysis of these two genes exhibited superior diagnostic performance.

PCA was performed to further validate their discriminatory power. The PCA plot, derived from dimensionality reduction of gene expression data in both HCs and SLE patients, demonstrated clear separation between the two groups based on the expression profiles of *FCER1A* and *RGS1* (Fig. 5). The first two principal components (PC1 and PC2)

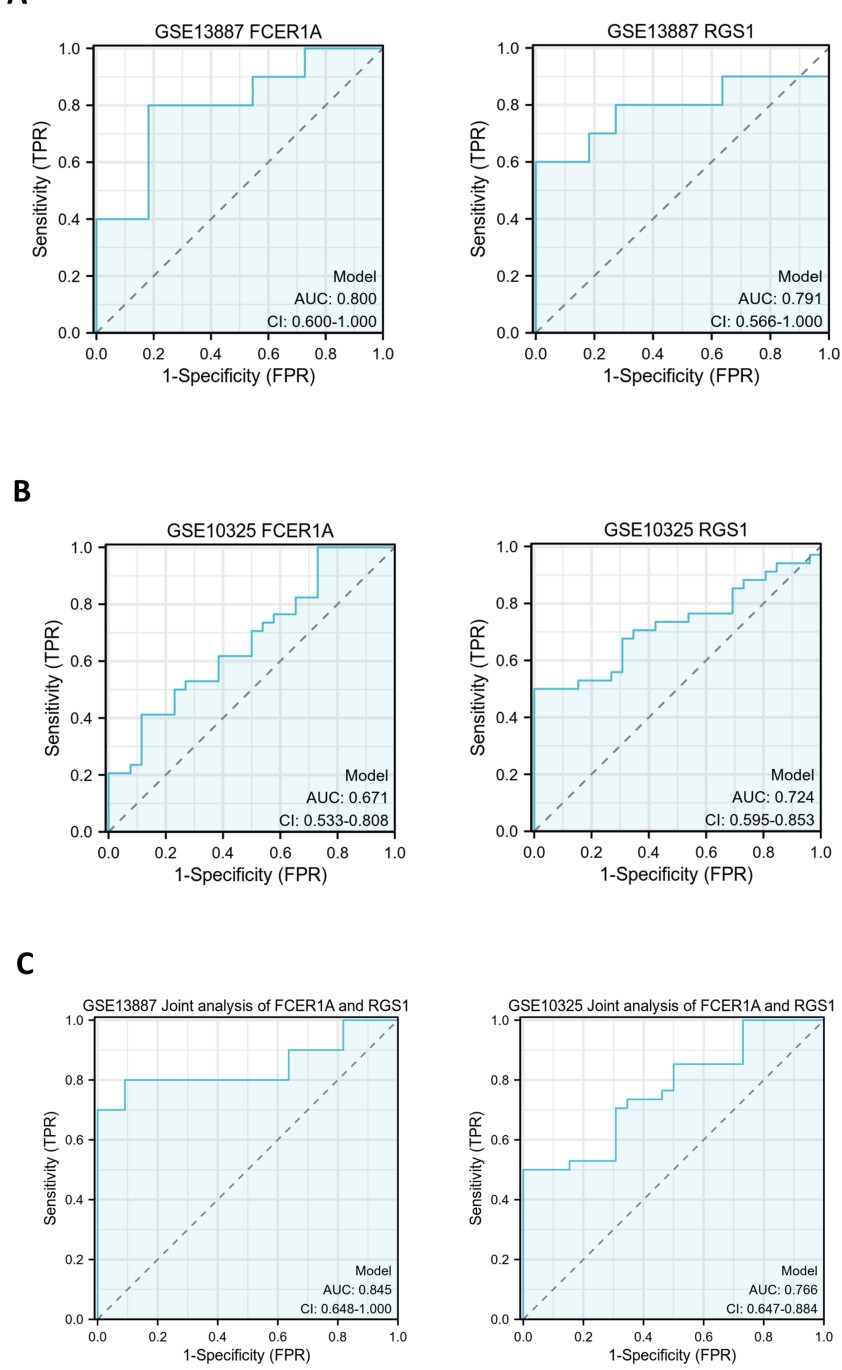

**Figure 4   Hub gene ROC curve.** (A) ROC curve analysis of FCER1A and RGS1 hub genes in the GSE13887 dataset. (B) ROC curve analysis of FCER1A and RGS1 hub genes in the GSE10325 dataset. (C) Joint ROC curve analysis of FCER1A and RGS1 genes in the GSE13887 and GSE10325 datasets.

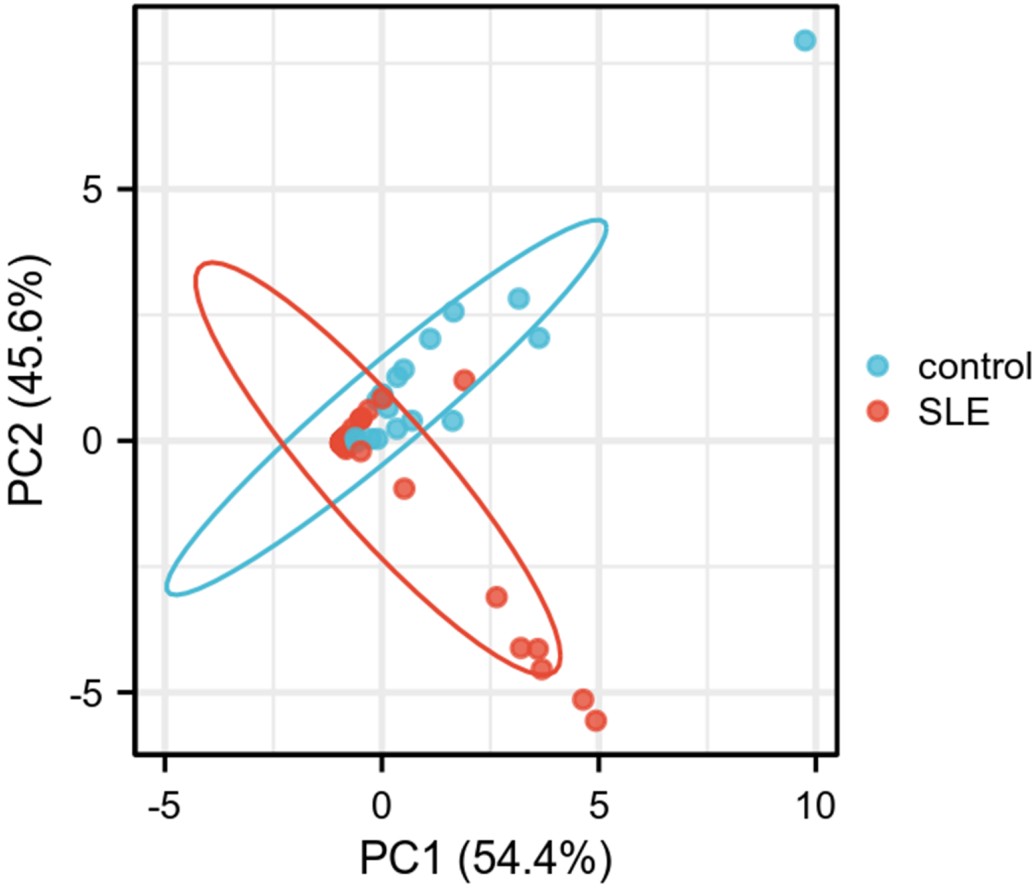

**Figure 5** **PCA plot of two genes.** PC1 and PC2 are the first and second principal components, respectively, explaining the different expressions of potential variants. The plots represent different samples, red for SLE group and blue for control group.

accounted for 100% of the explained variance. SLE samples clustered predominantly on the negative axis of PC1, while HCs were localized on the positive axis, indicating significant differences in gene expression patterns between the groups and highlighting the robust discriminatory capability of these two genes.

## Verification by other datasets and clinical samples

To further substantiate these findings, the expression of FCER1A and RGS1 was validated using the GSE61635 dataset, which corroborated their diagnostic efficacy. The expression trends observed in GSE61635 were consistent with those from GSE13887, GSE10325, and clinical samples, with *FCER1A* downregulated and *RGS1* upregulated in SLE patients relative to HCs (Fig. 6).

Peripheral blood samples were collected, and CD3$^+$ T cells were isolated using magnetic bead separation. RNA extraction followed by RT-qPCR revealed significant downregulation of FCER1A ($p < 0.0001$) and upregulation of RGS1 ($p < 0.0001$) in SLE patient samples compared to HCs (Fig. 7A). These findings were corroborated by flow cytometry, which

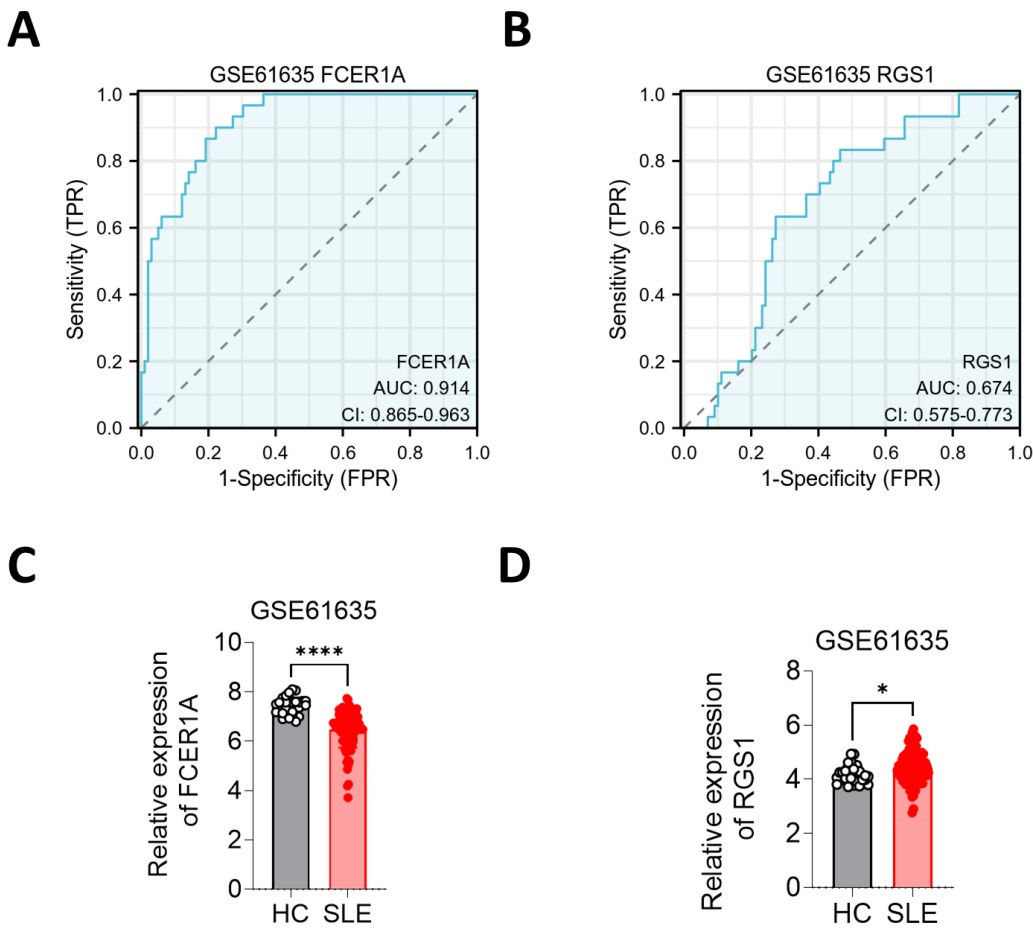

**Figure 6 Validation of FCER1A and RGS1 in other datasets.** (A, B) ROC curve analysis of FCER1A and RGS1 in GSE61635 dataset (C,D) Comparison of the expression of FCER1A and RGS1 in SLE patients and HCs Comparison between the two groups was performed using Mann–Whitney U test, $p < 0.05$ (*), $p < 0.01$ (**), $p < 0.001$ (***), $p < 0.0001$ (****), ns (not significant).

demonstrated a marked reduction in FCER1A protein levels ($p < 0.01$) and a significant increase in RGS1 protein expression ($p < 0.001$) in the plasma of SLE patients (Figs. 7B, 7C).

Additionally, flow cytometric analysis of inflammatory cytokines revealed elevated plasma levels of IL-6 and TNF-α ($p < 0.01$), alongside a significant reduction in IL-10 ($p < 0.001$) in SLE patients compared to HCs (Fig. 7D). These results further support the potential of FCER1A and RGS1 as robust biomarkers for the diagnosis and pathophysiological understanding of SLE.

## DISCUSSION

SLE is a multifactorial autoimmune disorder characterized by widespread inflammation and immune-mediated damage across multiple organ systems, with a notably higher prevalence in women (*Kaul et al., 2016*). The pathogenic mechanisms underlying SLE are intricate and

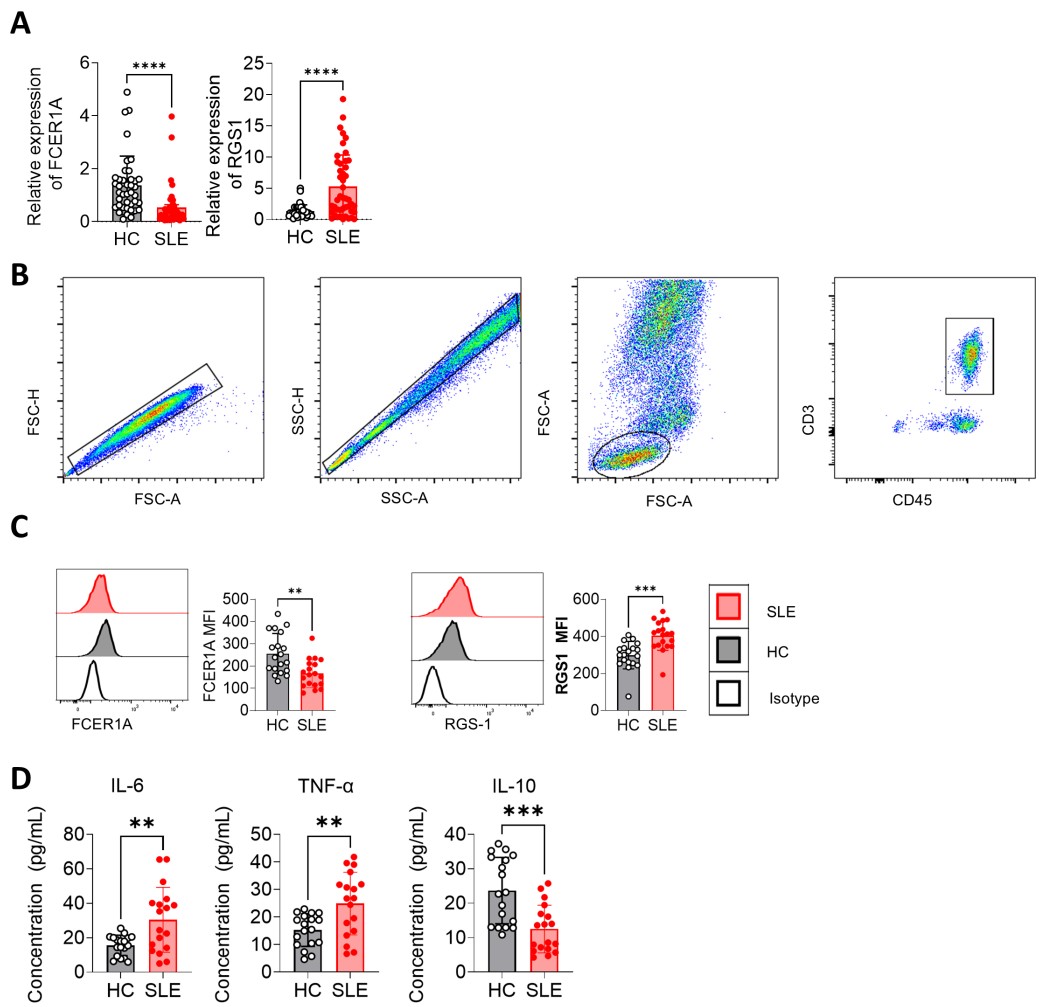

**Figure 7** **Validation of FCER1A and RGS1 expression in CD3 + T cells and assessment of inflammatory cytokines in plasma.** (A) RT- q PCR analysis of *FCER1A* and *RGS1* mRNA expression in CD3 + T cells from healthy controls and SLE patients. (B) Flow cytometry gating strategy. (C) Protein expression levels of FCER1A and RGS1 in CD3 + T cells from SLE patients and healthy controls, measured by flow cytometry. (D) Plasma levels of IL-6, TNF-α, and IL-10 in SLE patients and healthy controls. Statistical comparisons were performed using the Mann–Whitney U test: $p < 0.05$ (*), $p < 0.01$ (**), $p < 0.001$ (***), $p < 0.0001$ (****).

interdependent, encompassing T cell dysregulation, B cell hyperactivity, and an elevation in proinflammatory cytokines (*Parodis, Gatto & Sjöwall, 2022*; *Wangriatisak et al., 2022*; *Yap & Lai, 2013*). SLE manifests in a diverse array of clinical presentations and follows an unpredictable trajectory, complicating early diagnosis and therapeutic intervention (*Dörner & Furie, 2019*; *Durcan, O'Dwyer & Petri, 2019*). Hence, the identification of SLE-related biomarkers, genes, or signaling pathways is imperative for advancing diagnostic accuracy, therapeutic strategies, and prognostic evaluations in SLE.

Bioinformatics has emerged as a robust tool for predicting potential therapeutic targets and biomarkers in autoimmune diseases such as SLE. For instance, *Zhao et al. (2021)*

demonstrated through a comprehensive bioinformatics analysis that targeting *IFI27* holds promising therapeutic potential for SLE. *Shen et al. (2022)* identified a set of interferon-stimulated genes as potential diagnostic biomarkers for SLE. Leveraging transcriptomic data, *Wu et al. (2024)* established that the pathogenesis of COVID-19 shares certain similarities with that of SLE and screened small-molecule compounds, thereby suggesting potential molecular targets for the treatment of both COVID-19 and SLE in combination. *Li et al. (2023)* conducted an in-depth exploration of the molecular characteristics of NET-associated genes (NRGs) in SLE, identifying three prospective biomarkers (*HMGB1*, *ITGB2*, and *CREB5*), and categorizing three distinct clusters based on these key biomarkers.

In this study, we conducted an analysis of the GSE13887 and GSE10325 datasets associated with SLE using bioinformatics tools, and identified eight co-DEGs. This was followed by functional enrichment analysis. Among these, seven hub genes—*FCER1A*, *RGS1*, *CXCL13*, *DPEP2*, *LAG3*, *APOL1*, and *GZMB*—were extracted by constructing a PPI network, and subsequently validated in peripheral blood samples of SLE patients through RT-qPCR and flow cytometry. Upon evaluating the expression levels of genes across the two datasets, we identified two genes, *FCER1A* and *RGS1*, whose expression trends were consistent in both the datasets and clinical samples. *FCER1A*, a high-affinity IgE receptor, is significantly downregulated in SLE patients, suggesting potential impairment in antigen presentation and immune regulation (*Andiappan et al., 2021*; *Leffler et al., 2019*; *He et al., 2020*). Conversely, *RGS1*, a G protein signaling regulator, is notably upregulated, and its expression is associated with T cell migration and immune response modulation (*Jiang et al., 2024*; *Bai et al., 2021*). Functional enrichment analysis revealed that DEGs were significantly associated with biological processes such as "leukocyte-mediated cytotoxicity", "natural killer cell-mediated immunity", and "apoptosis". KEGG pathway analysis highlighted the enrichment of pathways related to "cancer transcriptional dysregulation", "apoptosis", and the "thyroid cancer pathway", underscoring the involvement of immune dysregulation and apoptosis in the pathogenesis of SLE. Previous studies have established that inflammation, the over activation of immune pathways, and compromised apoptotic mechanisms are central to the pathogenesis of SLE (*Crow, 2023*; *Sutanto & Yuliasih, 2023*; *Mistry & Kaplan, 2017*; *Mahajan, Herrmann & Munoz , 2016*). Our findings align with these reports, emphasizing the pivotal role of the hub genes identified within these pathological processes. Among the identified hub genes, *FCER1A* plays a critical role in immune regulation, participating in IgE-mediated signaling, while *RGS1* modulates G protein-coupled receptor signaling pathways, influencing immune cell migration and function. Likewise, *CXCL13* is known to regulate B cell trafficking and follicular helper T cell responses, both of which contribute to the pathogenesis of SLE (*Hui et al., 2024*; *Schiffer, Worthmann & Haller, 2015*). The serine protease *GZMB* has been implicated in cytotoxic T cell-mediated apoptosis, a crucial mechanism in immune homeostasis (*Thompson & Cao, 2024*). Additionally, *DPEP2*, *LAG3*, and *APOL1* exhibit well-established immunoregulatory effects, though their precise contributions to SLE progression remain to be fully characterized. Simultaneously, we assessed the levels of inflammatory mediators IL-6, TNF-α, and IL-10 in the collected plasma samples. It was observed that IL-6 and TNF-α levels were elevated, while IL-10 levels were diminished. IL-6, a proinflammatory

cytokine, plays a pivotal role in B-cell differentiation, T-cell activation, and the production of acute-phase proteins. The increased levels of IL-6 in SLE patients indicate its involvement in driving systemic inflammation, enhancing autoantibody production, and contributing to disease activity (*Karampetsou et al., 2019*; *Talaat et al., 2015*). TNF-α is a potent proinflammatory cytokine that plays a significant role in inflammation, tissue damage, and apoptosis in SLE. Elevated TNF-α levels are strongly associated with disease severity and organ damage (*Richter et al., 2023*; *Ghorbaninezhad et al., 2022*). IL-10, recognized for its anti-inflammatory properties, modulates excessive immune responses and ensures immune tolerance. Reduced IL-10 levels in SLE patients impair its protective function in suppressing inflammation and preventing immune dysregulation. A deficiency in IL-10 may foster the persistence of auto reactive B and T cells, thereby exacerbating autoimmunity (*Moore et al., 2001*; *Biswas, Bieber & Manz, 2022*).

Previous studies have made significant strides in identifying biomarkers for autoimmune diseases (ADs) in general. For instance, a review by *Gibson et al. (2010)* highlighted the challenges in diagnosing and predicting outcomes in autoimmune disorders, emphasizing the need for proteomic strategies to discover early biomarkers. The authors discussed the potential of proteomic platforms to reflect the complexity of autoimmune disease processes, suggesting that these approaches could lead to more accurate and timely diagnoses. Another study by *Kruta et al. (2024)* explored the application of machine learning for precision diagnostics of autoimmune diseases. The authors developed an integration pipeline to preprocess and integrate various types of health data, including clinical, laboratory, and multi-omics data, to improve the accuracy of machine learning models in classifying autoimmune diseases. Their results demonstrated that integrating multiple data types significantly enhanced the prediction accuracy, highlighting the potential of machine learning in personalized medicine for autoimmune conditions. A comprehensive review by *Vivas, Boumediene & Tobón (2024)* provided an overview of the latest advancements in predicting autoimmune diseases, including both traditional biomarkers and innovations in artificial intelligence. The authors discussed the potential of AI tools in predicting SLE, emphasizing the importance of early diagnosis and intervention. They also highlighted the need for further research to develop robust predictive models that can be applied in clinical settings. However, these studies summarized the advancements in biomarkers for ADs and outlined directions for future perspectives; in particular, these methods still require further investigation.

Despite these insights, our study still has some limitations. Although clinical samples provided preliminary validation, the functional role of the identified hub genes in the pathogenesis of SLE still needs to be further explored through *in vitro* experiments and animal models, and we are currently trying to establish a multicenter study for further validation.

In summary, our bioinformatics analysis identified key hub genes, including *FCER1A* and *RGS1*, which may serve as potential biomarkers and therapeutic targets for SLE. As research continues to unravel the complexities of SLE, the integration of genomic data with clinical findings will pave the way for improved diagnostic tools, personalized treatments, and better outcomes for SLE patients in the future.

## CONCLUSION

In this study, we employed integrative bioinformatics analysis and clinical validation to identify and verify *FCER1A* and *RGS1* as potential biomarkers for SLE. These genes exhibited consistent differential expression patterns across multiple independent datasets and patient-derived clinical specimens, with *FCER1A* markedly downregulated and *RGS1* significantly upregulated in individuals with SLE. Functional enrichment and PPI analyses implicated these genes in critical immunological processes, including immune regulation, cell-mediated cytotoxicity, and apoptosis—hallmarks of SLE pathogenesis. Moreover, dysregulated levels of pro- and anti-inflammatory cytokines in patient plasma further substantiate the immunopathological relevance of these findings. Collectively, our results provide a foundation for the potential application of *FCER1A* and *RGS1* in SLE diagnosis and suggest their promise as targets for future therapeutic strategies. Further mechanistic studies are warranted to elucidate their functional roles in disease progression.

**Abbreviations**

| | |
|---|---|
| ACR | American College of Rheumatology |
| ANA | antinuclear antibodies |
| anti-dsDNA | anti-double stranded DNA |
| AUC | area under the curve |
| BP | Biological Process |
| CC | Cellular Component |
| co-DEGs | co-differentially expressed genes |
| DEGs | differentially expressed genes |
| GO | Gene Ontology |
| HCs | healthy controls |
| HCT | hematocrit |
| HGB | hemoglobin |
| IFN-$\alpha$ | Interferon-$\alpha$ |
| IL | Interleukin |
| KEGG | Kyoto Encyclopedia of Genes and Genomes |
| LYM | lymphocyte |
| MF | Molecular Function |
| NET | neutrophil extracellular trap |
| NEU | neutrophil |
| PCA | principal component analysis |
| PLT | platelet |
| PPI | protein-protein interaction |
| RBC | red blood cell |
| ROC | receiver operating characteristic |
| RT- qPCR | Quantitative real-time PCR |
| SLE | Systemic lupus erythematosus |
| SLEDAI | SLE disease activity index |
| Sm | anti-Smith |
| Tregs | regulatory T cells |
| WBC | white blood cell |

## ACKNOWLEDGEMENTS

We would like to thank Professor Weinan Lai for participating in the discussion of this article and the diagnosis of SLE disease.

### Funding

This study was supported by the National Natural Science Foundation of China (82002218, 82202978), the Guangdong Basic and Applied Basic Research Foundation (2019A1515010103), the Guangzhou Basic and Applied Basic Research Foundation (2023A04J2359), the President's Fund of Nanfang Hospital, Southern Medical University (2022B041), the Southern Medical University College Students Innovation and Entrepreneurship Training Program (S202312121108), the Outstanding Youths Development Scheme of Nanfang Hospital, Southern Medical University (2024J007), and Guangdong Provincial Clinical Research Center for Laboratory Medicine (2023B110008). There was no additional external funding received for this study. The funders had no role in study design, data collection and analysis, decision to publish, or preparation of the manuscript.

### Grant Disclosures

The following grant information was disclosed by the authors:
National Natural Science Foundation of China: 82002218, 82202978.
Guangdong Basic and Applied Basic Research Foundation: 2019A1515010103.
Guangzhou Basic and Applied Basic Research Foundation: 2023A04J2359.
President's Fund of Nanfang Hospital, Southern Medical University: 2022B041.
Southern Medical University College Students Innovation and Entrepreneurship Training Program: S202312121108.
Outstanding Youths Development Scheme of Nanfang Hospital, Southern Medical University: 2024J007.
Guangdong Provincial Clinical Research Center for Laboratory Medicine: 2023B110008.

### Competing Interests

The authors declare there are no competing interests.

### Author Contributions

- Jie Xiao conceived and designed the experiments, performed the experiments, analyzed the data, prepared figures and/or tables, and approved the final draft.
- Yuhong Luo performed the experiments, analyzed the data, prepared figures and/or tables, and approved the final draft.
- Lina Duan performed the experiments, analyzed the data, prepared figures and/or tables, and approved the final draft.
- Xinru Mao conceived and designed the experiments, authored or reviewed drafts of the article, and approved the final draft.

- Lingyue Jin performed the experiments, analyzed the data, prepared figures and/or tables, and approved the final draft.
- Haifang Wang conceived and designed the experiments, authored or reviewed drafts of the article, and approved the final draft.
- Hongxia Wang conceived and designed the experiments, authored or reviewed drafts of the article, and approved the final draft.
- Jie Pan performed the experiments, authored or reviewed drafts of the article, and approved the final draft.
- Ying Gong conceived and designed the experiments, prepared figures and/or tables, authored or reviewed drafts of the article, and approved the final draft.
- Haixia Li conceived and designed the experiments, authored or reviewed drafts of the article, and approved the final draft.

## Human Ethics

The following information was supplied relating to ethical approvals (i.e., approving body and any reference numbers):

This study involves human participants and was approved by the Ethics Committee of Nanfang Hospital, Southern Medical University (NFEC-2025-026). Participants gave informed consent to participate in the study before taking part.

## Data Availability

Raw data is available in the Supplemental Files.

## Supplemental Information

Supplemental information for this article can be found online at http://dx.doi.org/10.7717/peerj.19891#supplemental-information.

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
