# Peer review of "Exploring differential gene expression and biomarker potential in systemic lupus erythematosus: a retrospective study"

_PeerJ, doi:10.7717/peerj.19891_

## Round 0.1 · original submission · Major Revisions

Please address the concerns of all reviewers and revise the manuscript accordingly. Please note that reviewer #1 indicated that the research design in your manuscript has a fundamental flaw. This is a serious issue that needs careful explanations.

Reviewer 1 ·

Basic reporting

-

Experimental design

-

Validity of the findings

The high-level logic of this manuscript does not hold.

The authors emphasized that current diagnosis methods for SLE are limited because "SLE is highly heterogeneous and may overlap with those of other autoimmune diseases".

In other words, the major current challenge is the difficulty in distinguishing SLE from other autoimmune diseases (ADs).

Motivated by this challenge, the authors analyzed the gene expression patterns between SLE and healthy controls, seeking to identify novel biomarkers that are of diagnostic use. This is where the primary problem is--to precisely classify SLE from other similar ADs, the data should come from SLE vs. other ADs, or firstly, healthy vs. autoimmune, then SLE vs. other ADs. Instead, the authors were still trying to classify between healthy controls and SLE, which does not address the problem they emphasized.

The secondary problem is that the sample size is too small for a reliable biomarker (n=32 vs. n=24) discovery study.

Reviewer 2 ·

Basic reporting

Xiao et al. reported a bioinformatics pipeline using two public datasets from the Gene Expression Omnibus (GEO) database that revealed a total of seven hub genes that could be used as potential biomarkers for systemic lupus erythematosus (SLE). The manuscript is well written in clear, professional English. A sufficient background was introduced. Conclusions are made with appropriate reference to relevant biological contexts. Experimental techniques including RT-PCR and Flow Cytometry was employed to validate computational findings. Overall, the article is satisfactorily self-contained.

Experimental design

The authors included a variety of bioinformatics methods to validate the proposed biomarkers for SLE, from differentially expressed gene calling to functional enrichment analysis, protein-protein interaction (PPI) network construction, ROC curve analysis, principal component analysis (PCA), and multiple visualization techniques. The questions are well defined and meaningful, closely orienting the proposed study goal. The investigation process is solid, satisfying rigorous scientific investigation practices.

The manuscript could be further improved if the authors could add more details to the following minor aspects of the study:

(1) The authors should discuss the specific disease stage or disease activities of the SLE patients sampled from the two GEO datasets. Empirically, different stages of the disease should manifest slightly different molecular genotypes.

(2) The authors should explain which DEG calling method was used, for example, if they used edgeR, DESeq2, or something else.

(3) The authors should explain which p-value correction method was used, if it is FDR, Bonferroni, etc.

(4) On Line 159, the authors mentioned that "both datasets were normalized." Please explain the normalization method.

Validity of the findings

This study is overall statistically sound and robust. Multiple potential confounding factors were considered and controlled. The conclusions are well written. The reviewer appreciates that the authors also provided an in-depth discussion of the biological relevance of the hub genes with regard to the SLE disease, supporting the potential clinical usefulness of the genes.

---

## Round 0.2 · Major Revisions

As you can see, one of the reviewers still has serious concerns about your study. Please address their queries and revise manuscript accordingly.

**PeerJ Staff Note**: Please ensure that all review, editorial, and staff comments are addressed in a response letter and that any edits or clarifications mentioned in the letter are also inserted into the revised manuscript where appropriate.

Reviewer 1 ·

Basic reporting

see "Validity of the findings"

Experimental design

see "Validity of the findings"

Validity of the findings

In the first round of review, I pointed out that this study is identifying biomarkers for distinguishing SLE vs. healthy controls, which is contradictory to the claimed challenge--distinguishing SLE vs. ADs.

Realizing this logical flaw, the authors have changed their objective to the following:

"our current findings serve as a basis for further investigation into distinguishing SLE from other autoimmune conditions. Our article only focuses on the diagnosis of SLE."

That means the objective of this study is diagnosing ADs in general (including SLE). However, there have been many studies discovering biomarkers for ADs, for examples, see the following reviews:

https://doi.org/10.1016/j.jprot.2009.11.013
https://doi.org/10.1038/s41598-024-76093-7
https://doi.org/10.1016/j.autrev.2024.103611

However, this manuscript did not explain how their study contribute to this field on top of the existing studies. Some of these studies are based on much larger sample size, and the small sample size is another weakness of this manuscript, as mentioned in the first round of review.

Reviewer 2 ·

Basic reporting

The authors have appropriately addressed all the comments previously made by this reviewer. There is no further comment.

Experimental design

no comment

Validity of the findings

no comment

---

## Round 0.3 · accepted · Accept

All remaining issues pointed out by the reviewer were adequately addressed, and the revised manuscript is acceptable now.